

# Modelling the underlying event in photon-initiated processes

Jonathan Butterworth[1], Ilkka Helenius[2,3], Juan Jose Juan Castella[4],
Bradley Pattengale[1], Shahzad Sanjrani[5,6] and Matthew Wing[1]

**1** University College London, London, UK
**2** University of Jyvaskyla, Department of Physics, P.O. Box 35,
FI-40014 University of Jyvaskyla, Finland
**3** Helsinki Institute of Physics, P.O. Box 64, FI-00014 University of Helsinki, Finland
**4** University of Cambridge, Cambridge, UK
**5** University of Bristol, Bristol, UK
**6** Deutsches Elektronen-Synchrotron DESY, Notkestr. 85, 22607 Hamburg, Germany

## Abstract

Modelling the underlying event in high-energy hadronic collisions is important for physics at colliders. This includes lepton colliders, where low-virtuality photons accompanying the lepton beam(s) may develop hadronic structure. Similarly, photon-induced collisions also occur in proton or heavy-ion beam experiments. While the underlying event in proton-proton collisions has been the subject of much study at the LHC, studies of hadronic-photon-induced underlying event are now of increasing interest in light of planned future lepton and lepton-hadron colliders, as well as the photon-induced processes in ultra-peripheral collisions at the LHC. Here we present an investigation of the underlying event in photon-initiated processes, starting from the Pythia models used to describe LHC and Tevatron data, and revisiting HERA and LEP2 data. While no single tune describes all the data with different beam configurations, we find that a good agreement can still be found within the same model by adjusting the relevant parameters separately for $\gamma\gamma$, $\gamma p$ and $pp$. This suggests that the basic model of multiparton interaction implemented in Pythia can be applied for different beam configurations. Furthermore, we find that a reasonable agreement for $\gamma\gamma$ and $\gamma p$ data, and for $pp$ data at an LHC reference energy, can be found within a single parametrization, but $pp$ collisions would prefer a stronger energy dependence, leading to too many multiparton interactions in lower energy photon-induced collisions. On this basis, we make some recommendations for simulations of photon-induced processes, such as $\gamma\gamma$ events at the LHC or FCC and $ep$ or $eA$ collisions at the EIC, and suggest possibilities for improvements in the modelling.


doi:10.21468/SciPostPhys.17.6.158

## 1  Introduction

It has long been understood that in modelling collisions of composite particles in which there are short-distance, high-momentum-transfer interactions between the constituents, the possibility that more than one pair of constituents undergoes such an interaction is phenomenologically significant. The first models of such a possibility were developed in the context of $Sp\bar{p}S$ data by Sjöstrand and van Zijl [1], and their descendants are implemented in the PYTHIA general-purpose event generators [2] and SHERPA [3,4]. Similar models were also developed in the context of photoproduction at HERA [5], with their descendants implemented in HERWIG [6].

All the above models and implementations have been widely used in proton-proton collisions at the LHC, having in most cases been previously tuned to data from the Tevatron [7]. The case of photon-induced processes was studied for the HERWIG and PYTHIA models several years ago [8] using HERA, LEP and TRISTAN data, and more recently the SHERPA implementation has been confronted with data from HERA and LEP [4]. Here we perform a similar study, with a somewhat wider range of data, for the current PYTHIA implementation. As with the SHERPA study, a prime motivation is the relevance of these models to upcoming data from the Electron Ion Collider [9]. We also note their relevance to photon-induced collisions at the LHC in both proton and heavy-ion running [10–12].

## 2  The PYTHIA model

The hard-process cross sections in PYTHIA are based on collinear factorization where the partonic structure of colliding hadrons, encoded in parton distribution functions (PDFs), are factorized from short-distance coefficient functions. Applying equivalent photon approximation (EPA) [13] one can also factor out the photon flux from the electron beam and sample the kinematics of the intermediate real photon based on the flux. By default the photon flux from charged leptons is given by

$$f_\gamma^l(x, Q^2) = \frac{\alpha_{\rm em}}{2\pi} \frac{1 + (1-x)^2}{x} \frac{1}{Q^2}, \tag{1}$$

but it is also possible to use a user-supplied photon flux.

The photon may interact either directly, or after developing a hadronic structure which is resolved in the collision. In the direct contribution, the photon scatters with a parton from the proton beam. In the resolved photon case, one needs to account for the partonic structure of (almost) real photons and sample a parton from the photon PDFs that participates in the hard process. In this case, the partonic evolution of the photon-proton subsystem proceeds as any hadronic collision including generation of multiparton interactions (MPIs) and parton-shower emissions. The possibility of MPIs is required to describe the data [14]. There are, however, a few differences that should be accounted for. First of all, the DGLAP evolution equation for

resolved photons includes a so-called anomalous contribution that describes the perturbative splittings of a photon to quark-antiquark pair. To remain consistent with the resolved-photon PDFs, a similar term has been included into initial-state radiation algorithm within the default parton shower in PYTHIA. Having this term in the backwards evolution allows a resolved-photon state to collapse back to an unresolved state during the partonic evolution. The parton showers are generated simultaneously with the MPIs and further interactions below the scale at which the photon has become unresolved are rejected. If the photon remains resolved until the minimum parton-shower emission scale, remnant partons are added to conserve colour and momentum.

The probability for MPIs in PYTHIA is obtained from the LO QCD $2 \rightarrow 2$ cross sections. These cross sections are regulated with the screening parameter $p_{\mathrm{T},0}$ which replaces the $1/p_{\mathrm{T}}^4$ behaviour by $1/(p_{\mathrm{T}}^2 + p_{\mathrm{T},0}^2)^2$ rendering the cross section finite also in the $p_{\mathrm{T}} \rightarrow 0$ limit. Having this parameter set, the average number of interactions can be calculated from $\langle n \rangle = \sigma_{\mathrm{int}}(p_{\mathrm{T},0})/\sigma_{\mathrm{nd}}$, where $\sigma_{\mathrm{int}}$ is the integrated cross section and $\sigma_{\mathrm{nd}}$ a parametrized cross section for non-diffractive scattering. In the initialization of the event generation the parameter is adjusted to a lower value if condition $\sigma_{\mathrm{int}}(p_{\mathrm{T},0}) > \sigma_{\mathrm{nd}}$ is not fulfilled. This ensures that there is always at least one partonic interaction in each non-diffractive collision event.

The rate of MPIs also depends on the overlap of the colliding particles in the impact-parameter space [15]. In the case of collisions with photons, one could expect a different distribution of the spatial overlap compared to proton-proton collisions. Here we, however, consider only the variation of $p_{\mathrm{T},0}$, and its energy dependence, that effectively accounts also for the modified spatial structure of the collision in the context of MPI generation. In addition, this spatial overlap is also affected by the hardness of primary scattering as discussed in the original article [1].

The parameter $p_{\mathrm{T},0}$ is dependent on the collision energy and is parametrized in terms of its value at the reference energy, $\sqrt{s^{\mathrm{ref}}}$, given by $p_{\mathrm{T},0}^{\mathrm{ref}}$ and the parameter $\alpha$ that controls the energy dependence. The scaling is either a power law (Eq. 2) or logarithmic (Eq. 3)

$$p_{\mathrm{T},0} = p_{\mathrm{T},0}^{\mathrm{ref}} \left( \frac{\sqrt{s}}{\sqrt{s^{\mathrm{ref}}}} \right)^{\alpha}, \tag{2}$$

$$p_{\mathrm{T},0} = p_{\mathrm{T},0}^{\mathrm{ref}} + \alpha \ln \frac{\sqrt{s}}{\sqrt{s^{\mathrm{ref}}}}. \tag{3}$$

The parameters $p_{\mathrm{T},0}^{\mathrm{ref}}$ and $\alpha$ can be tuned to experimental data after a suitable reference energy is selected. Current default values within PYTHIA for the values of the free parameters are given in Table 1. It is also possible to use a logarithmic scaling for the LHC and power-law scaling for LEP to test how the different extrapolations outside the fitted regions around $\sqrt{s^{\mathrm{ref}}}$ would compare to data. In this study we have used PYTHIA versions 8.308 and 8.310, which do not differ on the aspects relevant for the presented results.

Table 1: Default PYTHIA parameters for multi-parton interactions.

| Parameter | $pp$ | $\gamma\gamma$ |
|---|---|---|
| $p_{\mathrm{T},0}^{\mathrm{ref}}$ | 2.28 GeV | 1.54 GeV |
| $\sqrt{s^{\mathrm{ref}}}$ | 7000 GeV | 100 GeV |
| $\alpha$ | 0.215 | 0.413 |
| Scaling | Power | Logarithmic |

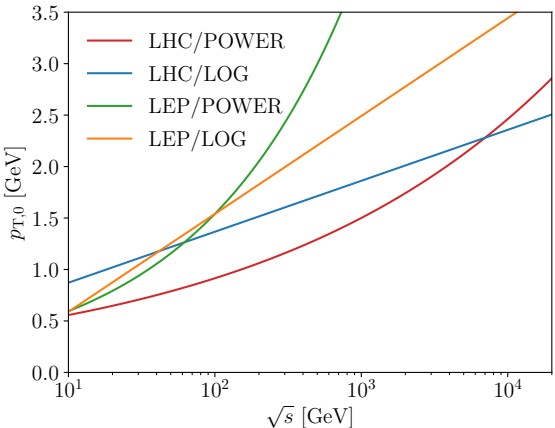

Figure 1: Energy dependence of $p_{T,0}$ with the considered parametrizations.

The following tunes were used in this study. The first four are based on the above parametrizations and values, and the final two consider hadron collisions of lower energy than the LHC. In the case of the latter two tunes also the parameters related to the overlap profile have been adjusted whereas with the four first tunes only the $p_{T,0}$ parametrization has been varied.

**LHC/POWER or Monash [16]** This tune is the default MPI tune for $pp$ and $ep$ collisions in PYTHIA. Its main focus is the description of LHC data, with some lower energy data used to tune the energy dependence.

**LHC/LOG** This is the same as the LHC/POWER tune with the scaling law changed to logarithmic with same value for $\alpha$ as above but now with a different role.

**LEP/LOG** This is the default parametrization in PYTHIA for photon-photon collisions [17] based on a tune to LEP data for charged particle production in $\gamma\gamma$ collisions [18] at $10 < W < 125$ GeV.

**LEP/POWER** This is the same as the LEP/LOG tune but with the power scaling law.

**Detroit [19]** This was developed to describe RHIC $p\bar{p}$ data at a centre-of-mass energy of 200 GeV, along with CDF data at centre-of-mass energies of 300, 900 and 1960 GeV. The Detroit tune uses newer parton density functions for the proton compared to the Monash tune.

**2C [20]** This tune predates the Monash tune and, like the Detroit tune, uses the lower energy CDF data.

The resulting energy dependency of $p_{T,0}$ is plotted in Fig. 1 for the four first parametrizations. By construction the LHC/LOG parametrization agrees with the LHC/POWER from the Monash tune at the reference energy of $\sqrt{s} = 7$ TeV, but interestingly gives similar values as the LEP/LOG tune around $\sqrt{s} = 100$ GeV relevant for the LEP and HERA comparisons. The LHC/POWER tune gives significantly lower values at these energies, resulting in a larger number of MPIs than with the other parametrizations considered.

# 3 Data for comparisons

For comparison of the parametrizations described above, we make use of RIVET [21, 22] version 3.1.10. RIVET provides implementations of many analyses from a wide variety of experiments, which are intended to be performed on the final state particles from simulated events. Histograms are produced which are directly comparable to the measurements, which RIVET obtains from HEPDATA [23].

In this study, measurements made in photon-initiated collisions are particularly relevant. Some were already available in RIVET, but several ($\gamma p[1, 3 - 5]$ and $\gamma\gamma 2$ below) have been newly implemented here and included in the RIVET codebase as an important resource for future studies.

### $\gamma p$ collisions

$\gamma p 1$ "Dijet cross-sections in photoproduction at HERA" by the ZEUS collaboration [24], RIVET ID ZEUS_1997_I450085. The two jets of highest transverse energy were both required to be above 6 GeV.

$\gamma p 2$ "Dijet photoproduction at HERA and the structure of the photon" by the ZEUS collaboration [25], RIVET ID ZEUS_2001_I568665. The two jets of highest transverse energy were required to be above 11 GeV, with the highest required to be above 14 GeV.

$\gamma p 3$ "Photoproduction of Dijets with High Transverse Momenta at HERA" by the H1 collaboration [26], RIVET ID H1_2006_I711847. The two jets of highest transverse energy were required to be above 15 GeV, with the highest required to be above 25 GeV.

$\gamma p 4$ "High-$E_T$ dijet photoproduction at HERA" by the ZEUS collaboration [27], RIVET ID ZEUS_2007_I753991. The two jets of highest transverse energy were required to be above 15 GeV, with the highest required to be above 20 GeV.

$\gamma p 5$ "Three- and four-jet final states in photoproduction at HERA" by the ZEUS collaboration [14], RIVET ID ZEUS_2007_I756660. Multijet photoproduction analysis from HERA for jets with $E_T^{\text{jet}} > 6$ GeV and $|\eta^{\text{jet}}| < 2.4$ in two different invariant mass bins of the multijet system.

$\gamma p 6$ "Charged particle cross sections in photoproduction and extraction of the gluon density in the photon" by the H1 collaboration [28], RIVET ID H1_1998_I477556. Charged particle tracks were required to have a transverse momentum above 2 GeV.

### $\gamma\gamma$ collisions

$\gamma\gamma 1$ "Di-Jet Production in Photon-Photon Collisions at $\sqrt{s_{\text{ee}}}$ from 189 GeV to 209 GeV", by the OPAL collaboration [29], RIVET ID OPAL_2003_I611415. Latest dijet measurement with the average of the two jets of highest transverse energy, $\overline{E}_T^{\text{jet}} > 5$ GeV.

$\gamma\gamma 2$ "Inclusive Production of Charged Hadrons in Photon-Photon Collisions", by the OPAL collaboration [18], RIVET ID OPAL_2007_I734955. Charged particle spectra with a minimum transverse momentum, $p_T > 1.5$ GeV and for different invariant masses, $W$.

### $pp$ collisions

$pp 1$ "Measurement of underlying event characteristics using charged particles in pp collisions at $\sqrt{s} = 900$ GeV and 7 TeV with the ATLAS detector" by the ATLAS collaboration [30], RIVET ID ATLAS_2010_I879407.

*pp*2 "Measurement of charged particle distributions sensitive to the underlying event in $\sqrt{s} = 13$ TeV proton-proton collisions with the ATLAS detector at the LHC" by the ATLAS collaboration [31], Rivet ID ATLAS_2017_I1509919.

## 3.1 Definition of kinematic variables

Most of the results considered here are measurements of jet cross sections, with common kinematic variables used in several analyses. The main kinematic variables are defined in this subsection.

Since the fraction of the photon's momentum, $x_\gamma$, participating in the hard scatter is not an observable, the variable $x_\gamma^{\text{obs}}$ was introduced at HERA [32] and also used at LEP. This approximates the fraction of photon's momentum participating in the production of the two or more jets of highest transverse energy. In the case of $n$ jets in the final state this quantity is defined as

$$x_\gamma^{\text{obs}} = \frac{\sum_{i=1}^{n} E_{\text{T},i}^{\text{jet}} e^{-\eta_i^{\text{jet}}}}{2 y E_e} , \tag{4}$$

where $y$ is the longitudinal momentum fraction of the almost-real photon emitted by the electron of energy $E_e$, $E_{\text{T},i}$ and $\eta_i^{\text{jet}}$ are the transverse energy and the pseudorapidity of the jet $i$ in the laboratory frame. In the case of dijet production at LEP a corresponding observable for each incoming photon (one with positive longitudinal momentum, another with negative longitudinal momentum) can be defined as

$$x_\gamma^\pm = \frac{\sum_{i=1}^{2}(E_i^{\text{jet}} \pm p_{z,i}^{\text{jet}})}{\sum_{j=1}^{n}(E_j \pm p_{z,j})} , \tag{5}$$

where the sum over $j$ includes all particles in the hadronic final state, excluding the scattered beam leptons.

The mean pseudorapidity of the two jets in the laboratory frame, $\bar{\eta}$, is given by

$$\bar{\eta} = \frac{\eta^{\text{jet1}} + \eta^{\text{jet2}}}{2} . \tag{6}$$

Note that at HERA, the positive $z$ axis is defined by the proton beam direction.

The absolute difference in azimuthal angle of the two jets, $\phi^{\text{jet1}}$ and $\phi^{\text{jet2}}$, is given by

$$|\Delta\phi| = |\phi^{\text{jet1}} - \phi^{\text{jet2}}| . \tag{7}$$

The dijet scattering angle, $\theta^*$, in the centre-of-mass frame is related to the two jets as follows:

$$\cos\theta^* = \tanh\left(\frac{\eta^{\text{jet1}} - \eta^{\text{jet2}}}{2}\right) . \tag{8}$$

For three-jet events, the angle $\psi_3$ was defined [14] to indicate the orientation of the third, lowest energy jet as follows:

$$\cos(\psi_3) = \frac{(\mathbf{p}_{\text{beam}} \times \mathbf{p}_3) \cdot (\mathbf{p}_4 \times \mathbf{p}_5)}{|\mathbf{p}_{\text{beam}} \times \mathbf{p}_3| \cdot |\mathbf{p}_4 \times \mathbf{p}_5|} . \tag{9}$$

Here $\mathbf{p}_{\text{beam}} = \mathbf{p}_p - \mathbf{p}_e$, where $\mathbf{p}_p$ and $\mathbf{p}_e$ are the three momenta of the proton and electron beams, respectively, and $\mathbf{p}_{3,4,5}$ are the three momenta of the the three jets in decreasing order of jet energy.

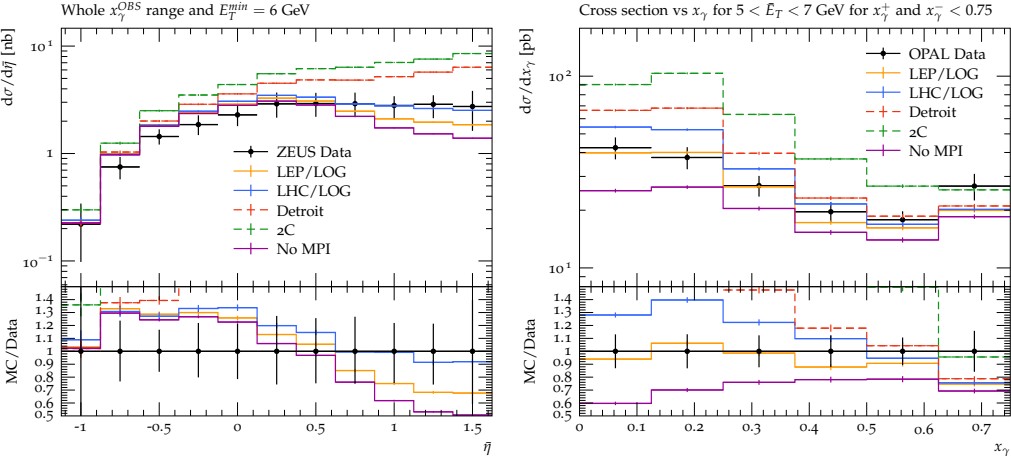

Figure 2: (Left) HERA photoproduction dijet data ($\gamma p1$) [24], where both jets are required to have transverse energies greater than 6 GeV, and (right) LEP dijet data ($\gamma\gamma1$) [29] compared to models of the underlying event in PYTHIA tuned to $\gamma\gamma$ data (LEP/LOG), LHC $pp$ data with modified energy dependence (LHC/LOG) and low-energy $pp$ data (Detroit, 2C), and to results without MPIs. The error bars on the PYTHIA expectations shown here and in subsequent figures represent statistical uncertainties.

## 4 Jet production data

A comparison to the lowest-transverse-energy jet data, available from ZEUS and OPAL, is shown in Fig. 2. It can be seen that while MPIs are required to describe the data, as mentioned previously, and the LEP and LHC tunes with logarithmic energy scaling both do a reasonable job, the Detroit and 2C models, tuned to describe proton-antiproton collisions at energies closer to the average photon-proton centre-of-mass energy at HERA and the average photon-photon energy at LEP2, do not model the data well; there is too much additional transverse energy in the events, enhancing the jet cross sections to the extent that they are far above the data, especially in regions where resolved photon interactions dominate – that is, at high rapidity at HERA (due to the asymmetric kinematics) and at low $x_\gamma^{\text{obs}}$ at LEP2.

In Fig. 3, comparisons to higher transverse energy jet data from ZEUS are shown. The conclusions regarding the Detroit and 2C tunes are the same as for Fig. 2. The LEP and LHC tunes with power law scaling are also shown, and it can be seen that of these, the LHC tune gives far too much activity at low $x_\gamma^{\text{obs}}$, while the LEP tune still gives a reasonable description. However, all the predictions overestimate the data in region around $0.6 < x_\gamma^{\text{obs}} < 0.8$ by $30-60\%$. Comparisons to the data in Fig. 3 were also made in Ref. [33], where the scale uncertainty on the PYTHIA prediction was estimated by varying the factorization and renormalization scales and found to be around 20%.

The H1 experiment also made measurements of $x_\gamma^{\text{obs}}$ in a very similar kinematic regime, which exhibit the same effects. In Fig. 4 we show the $\cos\theta^*$ distribution from these data in two regions of $x_\gamma^{\text{obs}}$. The excess at intermediate $x_\gamma^{\text{obs}}$ leads to the predictions lying above the data for the whole range of $\cos\theta^*$, since the mass cut made for this distribution enhances the relevant region of $x_\gamma^{\text{obs}}$. Also, due to invariant mass cut and large jet $E_T$ these data do not seem to be sensitive to MPI parameters, as LHC/POWER gives essentially the same cross sections as e.g. LEP/LOG with significantly higher $p_{T,0}$ at the relevant energies.

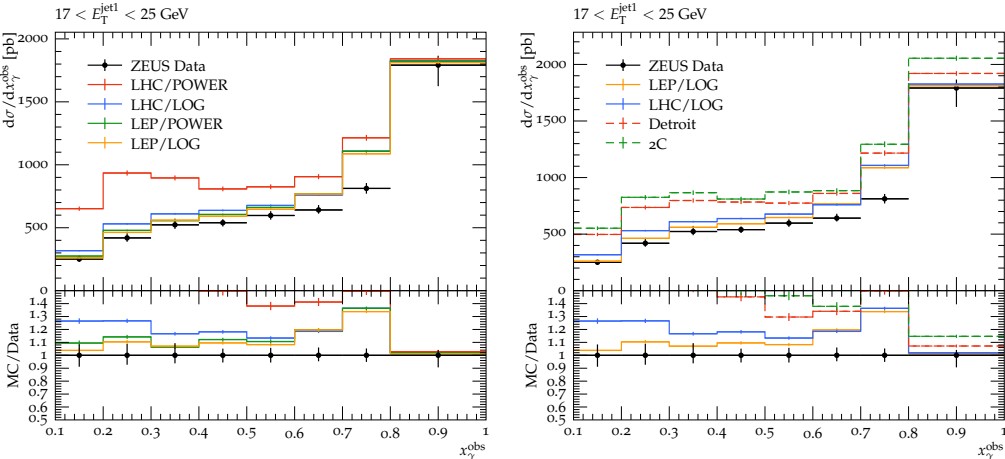

Figure 3: HERA photoproduction dijet data ($\gamma p2$) for the distribution $x_\gamma^{\mathrm{obs}}$, where the jet of highest transverse energy is required to be within the range $17 < E_{\mathrm{T}}^{\mathrm{jet1}} < 25$ GeV, compared to (left) default models of the underlying event in PYTHIA and (right) models tuned to lower energy data.

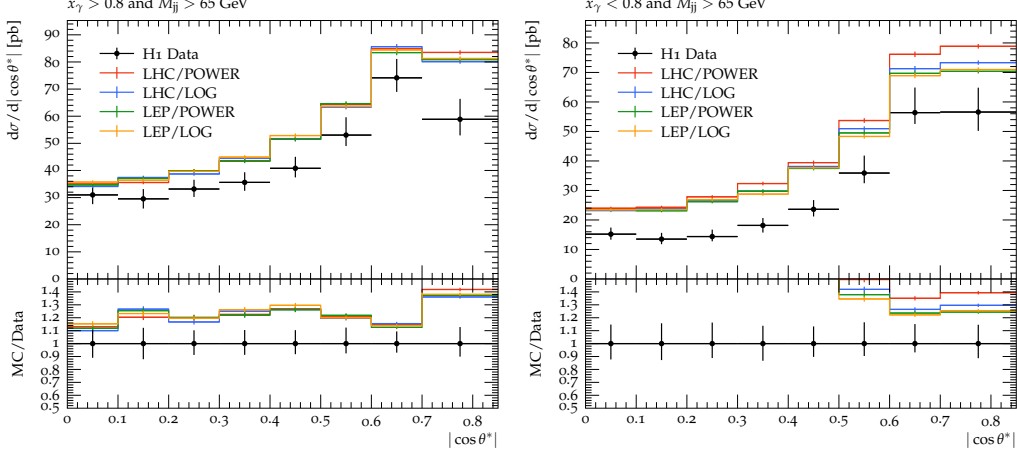

Figure 4: HERA photoproduction dijet data ($\gamma p3$) for the distribution $\cos\theta^*$, where the jet of highest transverse energy is required to be above 25 GeV and the dijet invariant mass above 65 GeV, shown for (left) $x_\gamma^{\mathrm{obs}} > 0.8$ and (right) $x_\gamma^{\mathrm{obs}} < 0.8$ compared to default models of the underlying event in PYTHIA.

At lower $x_\gamma^{\mathrm{obs}}$, the $\Delta\phi$ distribution, measured by the ZEUS collaboration [27], (see Fig. 5) is sensitive to hard QCD radiation; if the fraction of the photon's energy which is not present in the two highest $E_{\mathrm{T}}$ jets is collinear with the beam, it will not affect $\Delta\phi$, whereas extra high-$E_{\mathrm{T}}$ jets will lead to a broader $\Delta\phi$ distribution. The description is somewhat better than in the case of Fig. 4 and some sensitivity to $p_{\mathrm{T},0}$ parametrization can be found. Again, the LHC/POWER tune is clearly above the data at $\Delta\phi \approx \pi$ whereas other tunes are within 30% from the data.

Finally we compare to the multijet measurement from ZEUS [14] in Fig. 6. Multijet events can be formed either by producing several jets in a single hard partonic scattering, or by having several simultaneous partonic interactions that produce high-$p_{\mathrm{T}}$ partons. The former is approximately modelled by the parton-shower emissions and the latter with MPIs. Therefore, if the latter component is dominant, this observable can be very sensitive to the $p_{\mathrm{T},0}$ value. The sensitivity to MPIs is indeed significantly enhanced here, with simulation very far from the data when they are not included. The LHC power-scaled tune is also even more dramati-

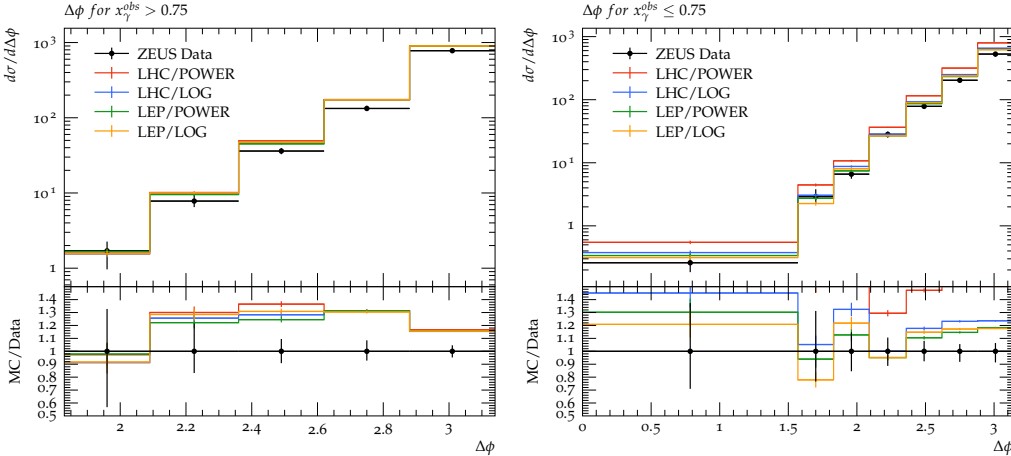

Figure 5: HERA photoproduction dijet data ($\gamma p4$) for the distribution of the difference in azimuthal angle of the two jets, $\Delta\phi$, where the jet of highest transverse energy is required to be above 20 GeV, shown for (left) $x_\gamma^{\text{obs}} > 0.75$ and (right) $x_\gamma^{\text{obs}} < 0.75$ compared to default models of the underlying event in PYTHIA.

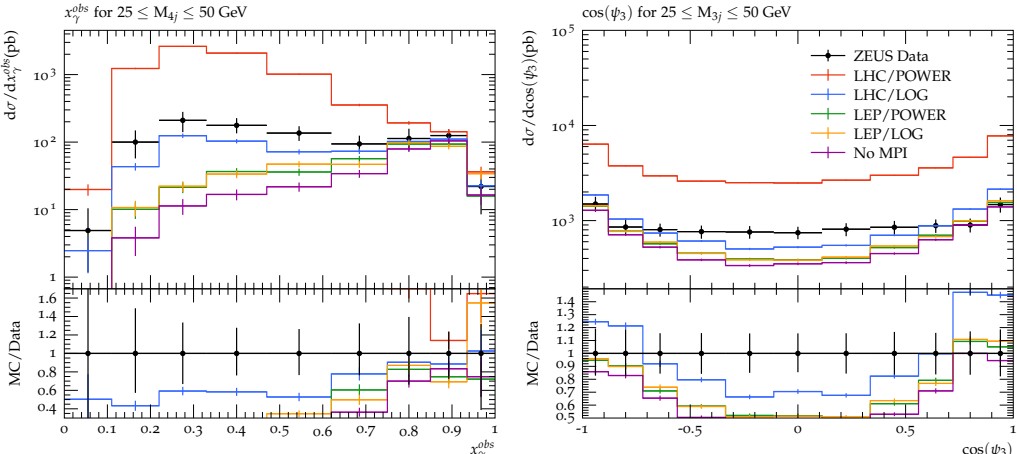

Figure 6: HERA photoproduction multi-jet data ($\gamma p5$) for the distributions (left) $x_\gamma^{\text{obs}}$ and (right) $\cos(\psi_3)$ (right). The $x_\gamma^{\text{obs}}$ distribution is for 4-jet events, and the $\cos(\psi_3)$ distribution is for 3-jet events, both requiring jets to satisfy $E_T^{\text{jet}} > 6$ GeV, $|\eta^{\text{jet}}| < 2.4$, and $25 \leq M_{nj} \leq 50$ GeV.

cally in conflict with the measurement. However, in this case it can be seen that both LEP tunes also fail badly at low $x_\gamma^{\text{obs}}$, with only the logarithmically-scaled LHC tune coming close to the data. We note that inclusion of higher-multiplicity matrix elements matched to the parton shower, which are not currently available in the simulation, would also be expected to impact these distributions. All tunes have a similar shape in the variable $\cos(\psi_3)$ and all differ to the data where the PYTHIA expectations have a steeper rise to $\cos(\psi_3)$ approaching $-1$ and $1$.

## 5 Charged hadron production

We now move away from jet production to charged particle production which is potentially more sensitive to the MPI tuning since these give access to lower values of $p_{\text{T}}$, though at the price of a less direct connection to hard-scattering kinematics.

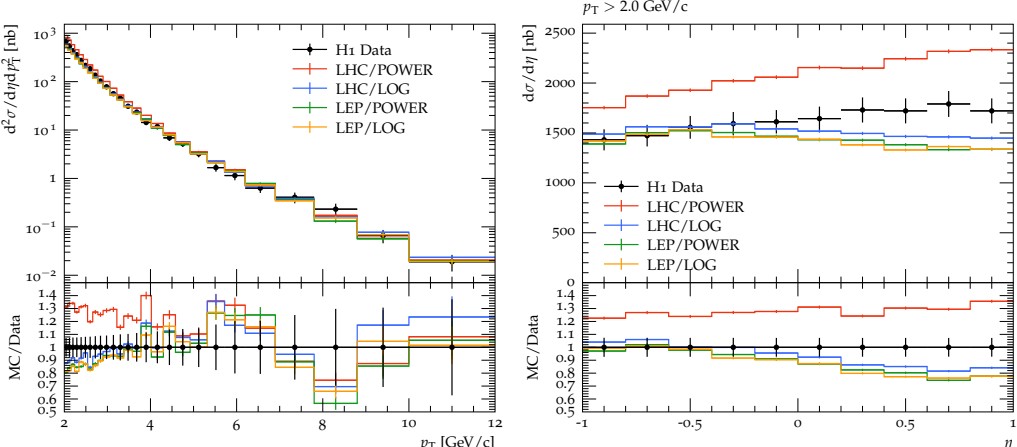

Figure 7: HERA photoproduction data ($\gamma p6$) for the production of charged hadrons with a transverse momentum above 2 GeV, shown for (left) transverse momentum, $p_T$, and (right) pseudorapidity, $\eta$, compared to default models of the underlying event in PYTHIA.

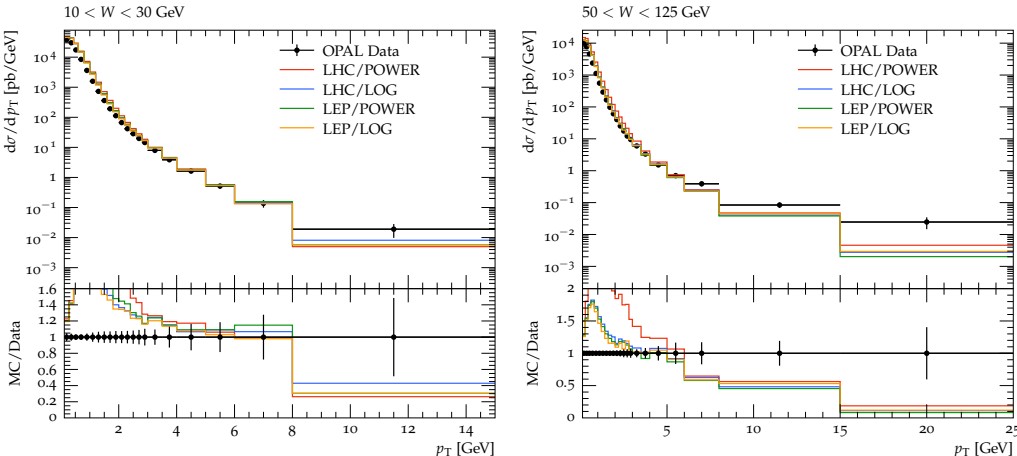

Figure 8: LEP2 $\gamma\gamma$ measurement ($\gamma\gamma 2$) of charged hadron production at different invariant mass bins of the $\gamma\gamma$ system.

Attempts have previously been made to tune the value of $p_{T,0}^{\text{ref}}$ to HERA [28] and LEP data [18]. The comparison with LEP data, a study [34] of charged hadron production yielded a value of about 3.3 GeV and in comparison to HERA data, a value of about 3.0 GeV was found to best describe the data. A complementary study using newer ZEUS inclusive charged particle data [35] also indicates that a value of about 3.0 GeV is preferred. These values are above the default used for LHC data of 2.28 GeV and indicate that fewer MPIs are present in photon-initiated processes, as was observed in the case of dijet production above.

The H1 measurement of charged hadron production is shown in Fig. 7 and compared to the default PYTHIA tunes, and in Fig. 8 the OPAL measurements from LEP2 are shown. In both cases the LHC/POWER tune gives too much activity, as already seen in the jet data, especially at low $p_T$. For the $\gamma p$ data, the other tunes give a good description for the central and backward regions, but fall below the data at positive rapidities. For the $\gamma\gamma$ data, the $p_T$ spectrum of particle is much too soft in both measured $W$ regions, with all models giving too many low $p_T$ particles and too few at high $p_T$.

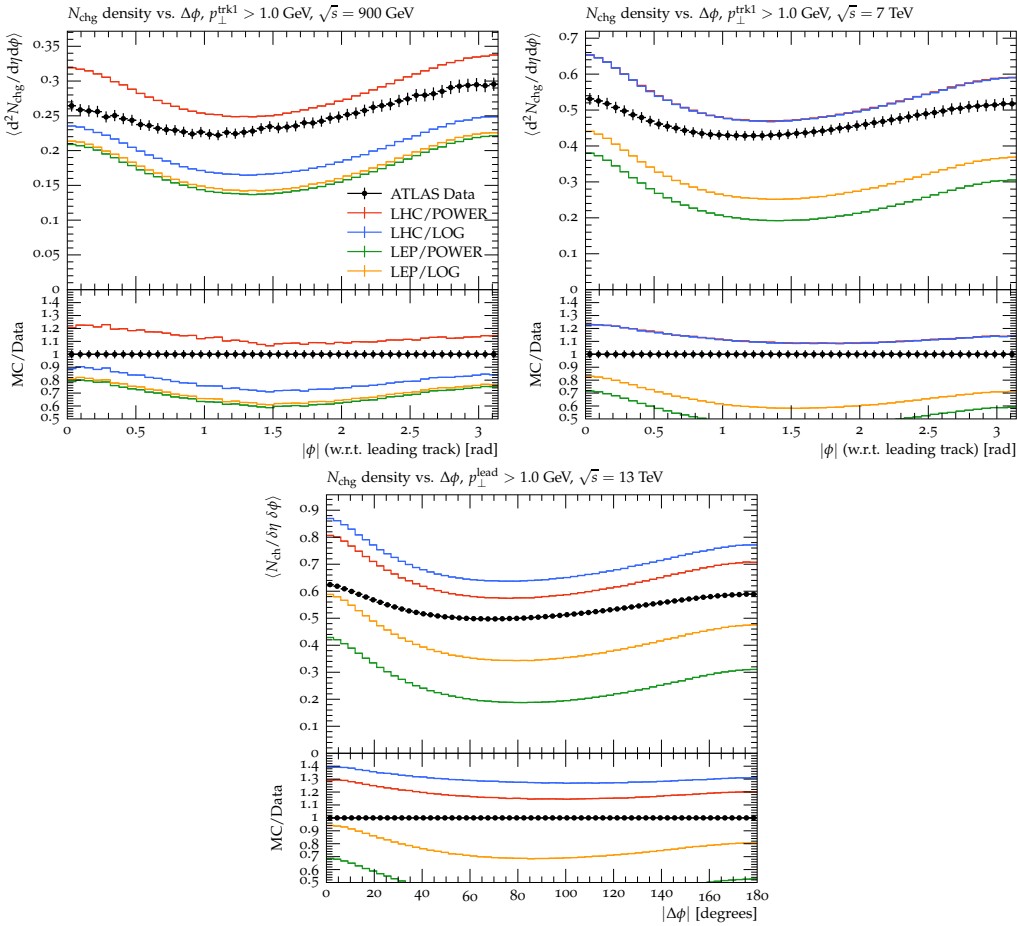

Figure 9:  ATLAS UE data for 900 GeV, 7 TeV (*pp*1), and 13 TeV (*pp*2).

Within the framework of the present study, the value of $\alpha$ was varied, with $p_{\mathrm{T},0}^{\mathrm{ref}}$ kept constant at 2.28 GeV, and compared to HERA and LEP data. The data preferred a value of $\alpha$ in the region of 0.05 and 0.1, significantly below the default of 0.215 used for LHC data, which resulted in an increased value for $p_{\mathrm{T},0}$ at energies relevant to LEP and HERA and therefore reduced MPI probability, further supporting observations in Figs. 7 and 8.

Finally we show these tunes compared to charged-hadron production in the underlying event at the LHC, for three different centre-of-mass collision energies – 900 GeV, 7 TeV [30], and 13 TeV [31] in Fig. 9. While none of the parameter settings describe the data perfectly the two LHC tunes are closer than the LEP tunes, which lie well below the data at all collision energies. We notice that the LHC/POWER tune retains approximately the same level of agreement with the data at all three collision energies, which is expected, since the tune made use of the lower-energy data as a constraint for the MPI parameters. The LHC/LOG tune moves from being below the data at 900 GeV to being above it at 13 TeV. This suggests that a power-law energy dependence is appropriate in proton-proton collisions. However, we have shown that this parameterization (LHC/POWER) gives poor agreement with the jet data from photon-induced collisions.

# 6 Discussion, recommendations and summary

Describing jet and charged particle production in high-energy colliders, especially at relatively low transverse momenta, is challenging for the Monte Carlo simulations commonly used in experimental design, in measurements and in interpretation of the data. Our study confirms that multiparton interaction models are an essential component of any reasonable description of these data, also in the case of processes initiated by low-virtuality photons.

Focussing on the PYTHIA model, comparisons to LEP2 and HERA data show that the default tune to proton-proton collisions at LHC energies does not describe these $\gamma\gamma$ and $\gamma p$ data at lower energy, but overshoots various observables by a large margin. In addition, we tested also tunes based on lower energy proton-(anti)proton collisions and found that also these tend to overestimate the measured cross section in photon-induced processes. However, if we switch the energy dependence of the key MPI parameter in the default Monash tune to be logarithmic instead of a power law, a reasonable agreement with both LEP2 and HERA data can be obtained. While this retains a reasonable agreement with the LHC data around the reference energy, the drawback is that the energy dependence of the LHC charged-multiplicity data does not come out right.

So far, tuning efforts for the new implementations of photoproduction and $\gamma\gamma$ collisions in MC event generators have been sparse. In the PYTHIA context, a default set of parameters for $\gamma\gamma$ collisions were derived based on LEP charged particle production data that can also fairly well describe the LEP jet production data, as shown in this study. However, this study also shows that applying this parametrization to HERA photoproduction data leads to an underestimate the cross sections for both charged particle and dijet production. This becomes especially evident in the case of multijet data from ZEUS, which seem to be very sensitive to the underlying MPI tune.

For modelling photon-induced events at present and future colliders with the current PYTHIA model, we would therefore suggest that the LHC/LOG tune should be used as a starting point for both $\gamma p$ and $\gamma\gamma$ collisions, since its description of the data is the most robust for changes of beam particle and energy in photon-induced collisions. For $pp$ collisions, the LHC/POWER tune is more likely to provide a reliable estimate of the energy dependence. There is, however, some amount of tension between HERA and LEP data that could motivate dedicated tunes separately for $\gamma\gamma$ and $\gamma p$ collisions. These observations and the new RIVET routines published within this study provide a good baseline for such future studies.

Another possible avenue would be to try out whether modifying the matter distribution, and therefore impact-parameter dependence of MPIs, for low-virtuality photons would allow the same parametrizations and energy dependence to be used in both proton-proton collisions and photon-induced collisions. Currently a Gaussian-like parametrization is applied for the overlap profile, but a form for the photon which is more peaked, for example based on the electromagnetic form factors [5, 36], is well-motivated and may influence the energy dependence of photon-induced events compared to those only involving hadrons.

In addition to MPI tuning, there are potentially other modelling aspects that could improve the description of the data with the current PYTHIA implementation. For example we notice that HERA $x_\gamma^{\mathrm{obs}}$ measurements in the $0.6 < x_\gamma^{\mathrm{obs}} < 0.8$ region and for high-$E_{\mathrm{T}}$ jets are poorly described in all parameter sets. Potentially this could be improved by re-fitting real-photon PDFs using more recent data and theory, especially from HERA. Explicitly including the possibility of a direct photon interaction with multiple partons may also have an impact [37]. Similarly, increasing the perturbative precision in the hard-process matrix elements and in the parton showers could potentially improve the description of the data in the regions where non-MPI related discrepancies occur.

# Acknowledgments

We are grateful to V.A. Narendran for providing a first version of the RIVET routine for analysis $\gamma p5$.

**Funding information** IH acknowledges the financial support from the Research Council of Finland, Project No. 331545, and through the Centre of Excellence in Quark Matter. MW acknowledges the support of DESY, Hamburg.

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
