# Peer review of "Modelling the underlying event in photon-initiated processes"

_SciPost Physics, doi:SciPost Phys. 17, 158 (2024)_

## Round 1 · Referee Report · Anonymous (Referee 1) · 2024-9-25

Strengths
1- important first step towards a consistent treatment of the underlying event/multi-parton interactions in preparation for the upcoming EIC and for photon physics at the LHC
2- timely
3- well written
Weaknesses
1- none
Report
This is an extremely well written clear paper with interesting results. It is important to pick up the issue of multi-parton interactions in processes with resolved photons, i.e. in processes, where the hadronic structure of the photon becomes experimentally visible and - quite often - dominant.
A first attempt to systematically scrutinise the Pythia model and to elucidate tensions between different data for processes with incoming protons and/or photons and parameters of the model will certainly pave the way for further studies. The observed tensions indicate a clear preference for some scaling laws etc. and it is not inconceivable that further studies may, possibly, highlight issues with the model itself.
I therefore recommend publication of the paper.
Requested changes
1- none
Recommendation
Publish (surpasses expectations and criteria for this Journal; among top 10%)

---

## Round 1 · Referee Report · Anonymous (Referee 3) · 2024-10-24

Strengths
This is an important contribution to a fast re-(emerging) field. It takes a fairly comprehensive approach to extending LEP and LHC Multi-Parton Interaction tunes to resolved photon processes in ep collisions at HERA, along with some gamma-gamma data from LEP. There are some clear conclusions in terms of which models are most successful. An important conclusion is that there is as yet no universal description of all processes and energies with a single model. Some pointers for future development towards that ultimate goal are given.
Weaknesses
Sometimes there's a lack of interpretation of results beyond statements of which models agree or and disagree with data. In an ideal world some comments might be made along the way, tied to the (good and appropriate) comments at the end of the discussion that there may be deficiencies in the photon matter distribution, PDFs or perturbative precision.
Report
The journals acceptance criteria are very clearly met. Please see other sections for more detailed remarks.
Requested changes
I do not insist on these changes, but encourage the authors to consider them.
1) It's not clear to me what the error bars that are sometimes visible on the MC predictions refer to. Presumably statistical?
2) Line 2 of section 4 asserts from the outset that MPI are required to describe the data. That seems quite strong to me on the evidence of fig 2, where the 'No MPI' model doesn't do too badly compared with some of the MPI models. Maybe it's fine if qualified by 'in the context of the PYTHIA model', but I guess from the high x_gamma region plots (4a, 5a) that PYTHIA with no MPI is not much better for direct photon processes than it is for fig 2. The best evidence for MPI probably comes from the 4 jet events (fig 6a) ... if only we had more of those data! Maybe the authors could consider a slightly more detailed discussion of whether MPI are present at all, possibly with the addition of 'No MPI' curves in figs 3-5?
3) In some MC/Data plots, the scale is such that the points are out of range (eg fig 4b). Maybe that could be changed?
4) For fig 2a, or in the definition of eta-bar on the previous page, it may be worth pointing out that this is in the lab frame and doesn't correspond to the gamma-p CMS.
Recommendation
Publish (easily meets expectations and criteria for this Journal; among top 50%)
Strengths
1- timely
2- written very clearly
3- conclusion with practical recommendations is very useful for the community and may serve as a good starting point for further study
Weaknesses
1- no real hint towards new developments of the modeling
Report
The paper is written very clearly and describes an important comparison to a wide range of available data involving photon induced hard processes. This is of interest to both the EIC community and further studies of ultra-peripheral hadron and ion collisions at the LHC. The authors describe the flexibility of the current Pythia model and recommend possible parameterizations of pt0 for the range of colliders and energies under study.
The conclusions could a bit point more towards future improvements of the modeling.
Recommendation
Publish (easily meets expectations and criteria for this Journal; among top 50%)

---

## Editorial Decision

published